# Occlusion and Deformation Handling Visual Tracking for UAV via Attention-Based Mask Generative Network

**Yashuo Bai** [1,2], **Yong Song** [1,2], **Yufei Zhao** [1,2,*], **Ya Zhou** [1,2], **Xiyan Wu** [1,2], **Yuxin He** [1,2], **Zishuo Zhang** [1,2], **Xin Yang** [1,2] and **Qun Hao** [1,2]

[1] School of Optics and Photonics, Beijing Institute of Technology, Beijing 100081, China
[2] Beijing Key Laboratory for Precision Optoelectronic Measurement Instrument and Technology, Beijing Institute of Technology, Beijing 100081, China
[*] Correspondence: zhaoyufei@mail.tsinghua.edu.cn

**Abstract:** Although the performance of unmanned aerial vehicle (UAV) tracking has benefited from the successful application of discriminative correlation filters (DCF) and convolutional neural networks (CNNs), UAV tracking under occlusion and deformation remains a challenge. The main dilemma is that challenging scenes, such as occlusion or deformation, are very complex and changeable, making it difficult to obtain training data covering all situations, resulting in trained networks that may be confused by new contexts that differ from historical information. Data-driven strategies are the main direction of current solutions, but gathering large-scale datasets with object instances under various occlusion and deformation conditions is difficult and lacks diversity. This paper proposes an attention-based mask generation network (AMGN) for UAV-specific tracking, which combines the attention mechanism and adversarial learning to improve the tracker's ability to handle occlusion and deformation. After the base CNN extracts the deep features of the candidate region, a series of masks are determined by the spatial attention module and sent to the generator, and the generator discards some features according to these masks to simulate the occlusion and deformation of the object, producing more hard positive samples. The discriminator seeks to distinguish these hard positive samples while guiding mask generation. Such adversarial learning can effectively complement occluded and deformable positive samples in the feature space, allowing to capture more robust features to distinguish objects from backgrounds. Comparative experiments show that our AMGN-based tracker achieves the highest area under curve (AUC) of 0.490 and 0.349, and the highest precision scores of 0.742 and 0.662, on the UAV123 tracking benchmark with partial and full occlusion attributes, respectively. It also achieves the highest AUC of 0.555 and the highest precision score of 0.797 on the DTB70 tracking benchmark with the deformation attribute. On the UAVDT tracking benchmark with the large occlusion attribute, it achieves the highest AUC of 0.407 and the highest precision score of 0.582.

**Keywords:** visual object tracking; unmanned aerial vehicle; adversarial learning; convolutional neural network; attention mechanism

## 1. Introduction

The main purpose of visual object tracking (VOT) [1] is to estimate the position and scale of the target in each subsequent frame in the videos, given the ground truth of the first frame. Meanwhile, the motion trajectory could also be well described. Therefore, it has been widely used in various fields, especially in unmanned aerial vehicle (UAV) applications, such as air surveillance [2], target following [3], and visual navigation [4]. Nevertheless, UAV-based remote sensing images and videos have intrinsic properties, such as image degradation, uneven object intensity, and small object size, that make UAV-specific tracking more challenging.

Recently, discriminative correlation filter-based (DCF-based) and convolutional neural network-based (CNN-based) trackers have made up the two streams of VOT methods.

Since the application of the correlation filter in object tracking [5], many outstanding DCF-based algorithms have been proposed with balanced accuracy and low cost for UAV tracking [6–8]. Meanwhile, CNN-based trackers, which are typically based on a two-stage tracking-by-detection framework, have achieved state-of-the-art performance in terms of accuracy and robustness [9–12]. Although the current VOT method has grown considerably, robust and accurate tracking for UAVs has remained a demanding task due to occlusion, deformation, illumination variation, and other challenges. Among various factors, occlusion and deformation are two of the main causes of tracking failure.

Various strategies have been proposed to address these challenges. The most intuitive paradigm is to build a network for occlusion and deformation and collect a large-scale dataset of the objects in different conditions to train the network, expecting to learn the invariance of object features eventually. For example, Zhou et al. [13] proposed a deep alignment network for multiperson tracking with occlusion and motion reasoning. A deep alignment network-based appearance model and a Kalman filter-based motion model were adopted to handle the occlusion. Wu et al. [14] combined an adaptive Kalman filter with a Siamese region proposal network to make full use of the object's spatial–temporal information, thereby robustly dealing with complex tracking scenes, such as occlusion or deformation. Yuan et al. [15] adopted ResNet to extract more robust features, in which the response maps computed from ResNet were weighted and fused using to realize accurate localization during tracking under various conditions.

However, the occlusion and deformation always follow a long-tail distribution, some of which are rare or even nonexistent in large-scale datasets [16]. Therefore, learning invariance to such rare/uncommon occlusions and deformations needs to be addressed urgently. To alleviate this problem, one way is to dealing with different challenging situations without requiring more training samples by designing different coping strategies specifically for different situations. For example, ref. [17] designed an attribute-based CNN with multiple branches, each of which is used to classify objects with specific attributes, thereby reducing the diversity of object appearance under each challenge and reducing the demand for the amount of training data. Ref. [18] adaptively utilized level set image segmentation and bounding box regression techniques to deal with the deformation problem, while designing a CNN to classify objects as occluded or non-occluded during tracking, thereby avoiding collecting samples updated by the occlusion tracker. These methods achieved robust and accurate tracking in a variety of complex situations without requiring a larger sample size but may not be sufficient in the face of more complex and variable situations.

Another method is to enrich the expressive power of samples for different challenge scenarios without requiring more actual samples. Considering the advantages of generative adversarial networks (GANs) in sample generation, many works adopted GANs to increase the diversity of training samples, thereby improving the tracker's ability to cope with challenges, such as occlusion and deformation. Wang et al. [16] proposed to adopt the adversarial network to enrich data samples with occlusion and deformation. This approach essentially generates samples that are difficult to be classified by the target detector, driving the adversarial system to produce a better detection network. Chen et al. [19] further introduced GANs into the problem of face detection and proposed an adversarial occlusion-aware face detector (AOFD). The role of the generative model in the algorithm is also to cover the key features of the face by generating masks in the training set. Likewise, to increase positive samples, Song et al. [10] employed the generation network to generate masks randomly, which adaptively discarded the input features to capture various appearance changes. After the adversary learning, the network can identify the masks that maintain the most robust features of the target object for a long time. Similar thinking was utilized by Javanmardi [20] to reduce the influence of object deformation on tracking and detection. In image space, Souly et al. [21] developed a semi-supervised semantic segmentation approach, which employs GANs to generate plausible synthetic images, supporting the discriminator in the pixel-classification step. Differently, Wang et al. [22] skillfully combined the application in the image and feature space of GANs to further

supplement the hard positive samples by using part of the image background to cover the target.

Like other methods partwise modeling object appearance [23], adversarial learning methods devote efforts to concentrate the classification network into some other features besides the visible parts of a target, which are more robust for giving reliable cues for tracking when the target is occluded and deformable. How to distinguish these features is the key. In the processes mentioned above, GAN predicts masks with $3 \times 3$ size to respectively cover the part of feature maps and dropout to adversarial training for the object tracking without these local features. Nevertheless, this mask is updated to cover only a portion of the features to select local features but is actually not enough to simulate occlusion and deformation. At the same time, inevitably, feature loss may make tracking drift in extreme situations, which here refers specifically to target occlusion and deformation, and the $3 \times 3$ feature maps from CNN contain less location and shape information of the object, which cannot give the object a thorough description. In this paper, we propose an attention-based mask generative network-based tracker, which we call the AMGN-based tracker, to address the above issues. The main contributions can be concluded as follows:

1. We propose an attention-based mask generative network-based (AMGN-based) tracker. First, we adopt a base deep CNN to extract the deep features of the candidate regions. Next, we use AMGN to generate a series of attention-based masks, which are applied to the deep feature to augment hard positive samples. Then, we design a feature fusion method to compensate for the possible over-subtraction of the features of hard positive samples by the masks and to compensate for target location information. Finally, these hard positive samples are used for subsequent generative adversarial learning, thereby improving the ability of the tracker to handle occlusion and deformation.

2. We develop an attention-based mask generative network (AMGN). After CNN extracts the deep features of the candidate region of which the salient positions are obtained through the attention module, masks for occluding the corresponding positions are generated. Multiply these masks with the deep features to simulate target occlusion and deformation in the feature space.

3. We design a feature fusion method. When multiplying the masks with the deep features, some features are discarded, and there is a chance that too many features are discarded in the process. To alleviate this problem, we incorporate shallower-layer features into deeper-layer features processed by masks, thus avoiding extreme cases of tracking drift due to excessive feature loss.

After the process of AMGN and feature fusion, many hard positive samples are generated. The enhancement of the occlusion and deformation training samples strengthens the object-tracking ability when the target is occluded and deformable by effectively covering the distinguishable features of the object and conducting confrontation training with the classification tracker as the discriminator. As a result, even if the target is occluded and deformable, the features of the unobstructed area assist target tracking. Figure 1 presents the principle of our method selecting local features and generating masks.

The rest of the paper is organized as follows. Section 2 covers related work. Section 3 describes the proposed method for tracking the occluded and deformed object, including the overall pipeline, base deep CNN, AMGN, and feature fusion method. Section 4 presents the comparison experiments and ablation studies. Finally, conclusive remarks and future research directions are given in Section 5.

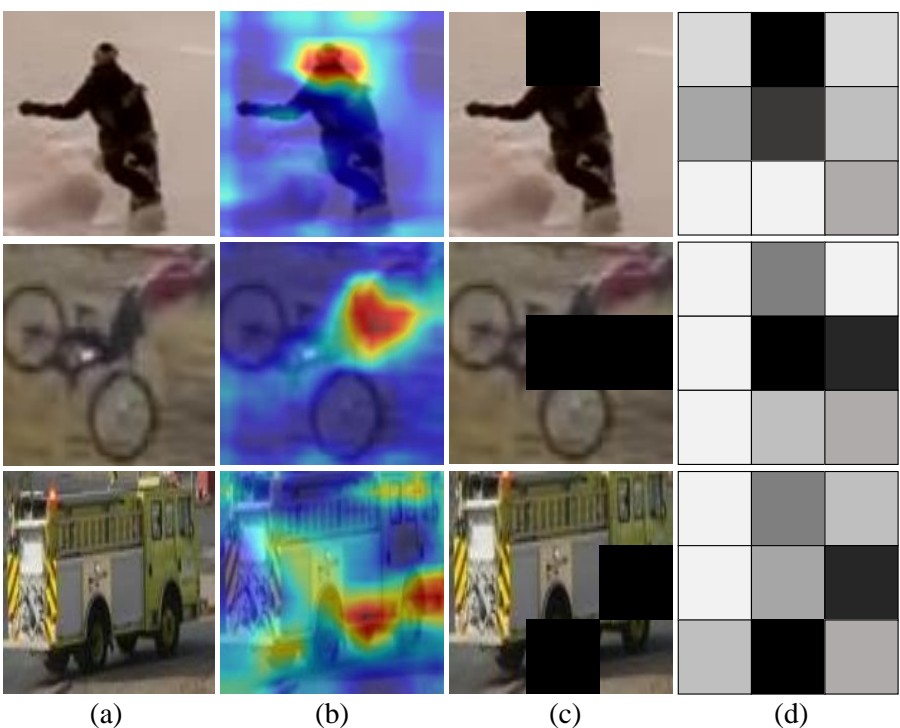

(a)          (b)          (c)          (d)

**Figure 1.** Principle of local feature selection and mask generation. In this paper, we propose to use an attention module and adversarial network to generate examples with occlusions and deformations that will be hard for the object tracker. The attention map is the visualization results of Grad-CAM [24] that learn the spatial attention of the target region. (**a**) Target region. (**b**) Attention map. (**c**) Mask template. (**d**) Generated mask.

## 2. Related Work

### 2.1. Occlusion and Deformation Handling in Visual Tracking

In general, visual tracking methods can be categorized as generative and discriminative. Generative methods extract target features before tracking to establish an appearance model that can represent the target. The model is then applied to pattern match the entire image and locate the most similar region. Typical generative model tracking algorithms include tracking algorithms based on Kalman filter [25], particle filter [26,27] and mean shift [28]. The generative methods only focus on tracking the target itself but ignore the background information, which is prone to tracking drift when the target is occluded or deforms drastically. The discriminative methods based on various approaches ranging from the traditional correlation filter (CF) [29,30], support vector machine (SVM) [31] to the currently widely used convolutional neural networks (CNNs) [32,33], GANs [10,34], recurrent neural networks (RNNs) [35,36], and especially Siamese neural networks [11,37] and other costume neural networks [38,39], always generate multiple suggestion boxes at first and then categorize each suggestion box into the target or background, employing offline pre-training and online learning. Nowadays, deep detection tracking methods, including multi-domain learning, ensemble learning, adversarial learning, reciprocating learning, and overlap maximization, have gradually become the mainstream of target tracking research due to various online update detector models, as they can better adapt to the complex changes of target objects in the tracking process.

Object tracking becomes a challenge when dealing with occluded and deformable objects, as they receive an incomplete description that does not resemble the patterns stored initially. Even if the selected candidate is indeed the target, the similarity between the features of the candidate image and the target image will not reach the threshold due to the effects of occlusion and deformation. Furthermore, the viability of deep learning relies on massive amounts of training data. When faced with the target occlusion and

deformation problem, if the positive occlusion and deformation samples on each frame are highly overlapping, it is difficult for the deep learning model to capture the target features under large-scale occlusion and deformation.

To handle occlusion and deformation robustly, several strategies [40–44] have been used. In deep learning methods, data collection and annotation is the most straightforward way, while it seems impossible to collect data covering all potential occlusion and deformation, even for large-scale datasets. Considering the advantages of GAN in data generation, numerous works have attempted to use GAN to generate occlusion and deformation images that meet the requirements. In addition, modal segmentation is another approach to reducing the existence, degree, and contours of occlusion and deformation by exploiting its ability to infer the physical structure of objects. By the way, the modal training data are created by adding synthetic occlusion and deformation to the modal mask. As with conventional methods, it is also popular to divide the target image or region of interest into some cells or segments, and then analyze each segment individually to improve the accuracy of the tracking model. For example, Zhan et al. built a self-supervised framework for partially completed occluded objects for scene de-occlusion. Pathak et al. proposed a CNN that can generate missing paths of an image based on context. Nonetheless, human beings have a remarkable ability to detect and recognize objects when they are partially visible and deformable. Some human vision mechanisms are introduced to learn appropriate attention parameters in different channels and effectively handle different occlusion and deformation patterns [45,46]. Among various human vision mechanisms, the attention mechanism has shown to be effective in many computer vision tasks, for which we will make a brief review in the next subsection.

### 2.2. Attentional Mechanisms in Neural Networks

We aim to learn more robust target appearance models with the help of spatial and temporal attention. Informally, the neural attention mechanism enables a neural network to focus on a subset of its inputs (or features), i.e., it selects specific inputs. Let $\mathbf{x} \in \mathcal{R}^d$ be the input vector, $\mathbf{z} \in \mathcal{R}^k$ be the feature vector, $\mathbf{a} \in [\mathbf{0}, \mathbf{1}]^k$ be the attention vector, $\mathbf{g} \in \mathcal{R}^d$ be the attention glimpse, and $f_\phi(\mathbf{x})$ be the attention network with parameters $\phi$. Typically, attention is implemented as

$$\mathbf{a} = f_\phi(\mathbf{x}), \tag{1}$$

$$\mathbf{g} = \mathbf{a} \odot \mathbf{z}, \tag{2}$$

where $\odot$ is the element-wise multiplication, and $\mathbf{z}$ is the output of another neural network $f_\theta(\mathbf{x})$ with parameters $\theta$. In this case, the attention mechanism introduces multiplicative interactions into the neural network space, making it simple and compact. Taking matrix-valued images as an example, most of the research on the combination of deep learning and visual attention mechanism focuses on using masks to achieve an attention mechanism, identifying key features in images through another layer of weights, and learning what needs to be paid attention to, thereby forming attention. This idea has evolved into soft attention and hard attention. Relatively, soft attention is more applicable in the task of object tracking to obtain alignment weights [47].

Soft attention, attaching importance to the spatial scales and channel scales, could be explicitly determined through network learning. Moreover, its differentiable characteristic allows neural networks to calculate gradients and learn the weights of attention by forwarding propagation and backward feedback. Among them, SENet channel attention [48] is to allocate resources between each convolutional channel and selectively enhance the features with the largest amount of information so that subsequent processing can make full use of these features and suppress useless features. The residual attention network for image classification combines the attention of the spatial domain and the channel domain while combining the ideas of the residual network of ResNet. Subsequently, problems, such as rare information retained after mask processing and the difficulty of stacking deep network structures, would be well prevented. Based on SENet, CBAM [49] consists of two

independent sub-modules, channel attention module (CAM) and spatial attention module (SAM), which realize channel attention and spatial attention, respectively. As a lightweight general-purpose module, it can be seamlessly integrated into any CNN architecture without the overhead and can be trained end-to-end with a basic CNN.

During VOT, the frequent disappearance, reappearance, and deformation of objects arouse tracking failures. Adopting an extra attention module can generate feature weights to select features and enhance the ability of feature expression. Combined with generative adversarial learning, the invariance of these important features can be effectively learned, thereby effectively improving the performance of target tracking algorithms.

### 2.3. Generative Adversarial Learning

GANs [50] have emerged as one of the hottest research fields in deep learning since they were proposed by Goodfellow et al. in 2014. Under the guidance of zero-sum game theory, the idea of a confrontation game runs through the whole training process of a GAN. It not only brings excellent generation quality to the model itself, but also is integrated into a series of traditional methods, forming a large number of new research directions. In terms of sample generation, the essence of GAN is a concept generation model, that is, to find out the statistical rules within a given observation data and generate new data similar to the observation data based on the probability distribution model obtained. On the other hand, GAN cleverly combines (self-)supervised learning and unsupervised learning, providing a new method for sample generation.

During the target tracking, the online training samples are not available before occlusion emerges. As a result, the tracking drift happens when the target is repeatedly blocked, and deforming for the tracking model is absent of the corresponding processing capacity. To tackle this problem, one solution is to furnish occlusion samples according to image synthesization. At present, there is a great deal of research work on image generation (pixel level) in various image generation algorithms [51]. Image generation technology based on the generative adversarial network has been able to generate real-like sample images with guaranteed quantity and diversity according to various requirements. Compared with other image generation networks, the generative adversarial network has lower complexity and higher flexibility. However, even if the sample images with occlusion can be supplemented in this way, it is still an arduous operation to provide the sample image with target tracking under complex background. A larger image sample database also meets the same problem for the long-tail problem and still has non-scalability.

In order to reduce the difficulty of sample replenishment, another solution is to add positive samples in the feature space to capture the appearance changes of the target in the time domain so as to improve the ability of the model to resist occlusion. Due to the flexibility of GAN, the training framework based on a generative adversarial mechanism can be combined with various types of loss functions according to specific tasks, and any differentiable function can be used as a generator and a discriminator. This way, there is no need to collect occlusion samples as the training base or consider the realistic rationality of generating samples, but there is a greater increase in the number of samples also containing as many diversities and features of occlusion.Consequently, the classification network, as a criterion, has stronger robustness in the process of confronting the generator network. It chiefly saves manpower, material, and financial resources.

## 3. Method

### 3.1. Overview

The proposed anti-occlusion and anti-deformation AMGN-based tracker consists of three modules. Firstly, the feature extraction of candidate regions is carried out through a base deep CNN. After that, AMGN utilizes feature maps and spatial attention weights of candidate regions to generate hard positive samples. Finally, the discriminant network distinguishes whether the features belong to the target or the background according to the fusion of the second and third convolution layer's features. Consequently, the tracking

model captures the anti-occlusion and anti-deformation ability. Figure 2 shows the pipeline of our method, and the details are discussed below.

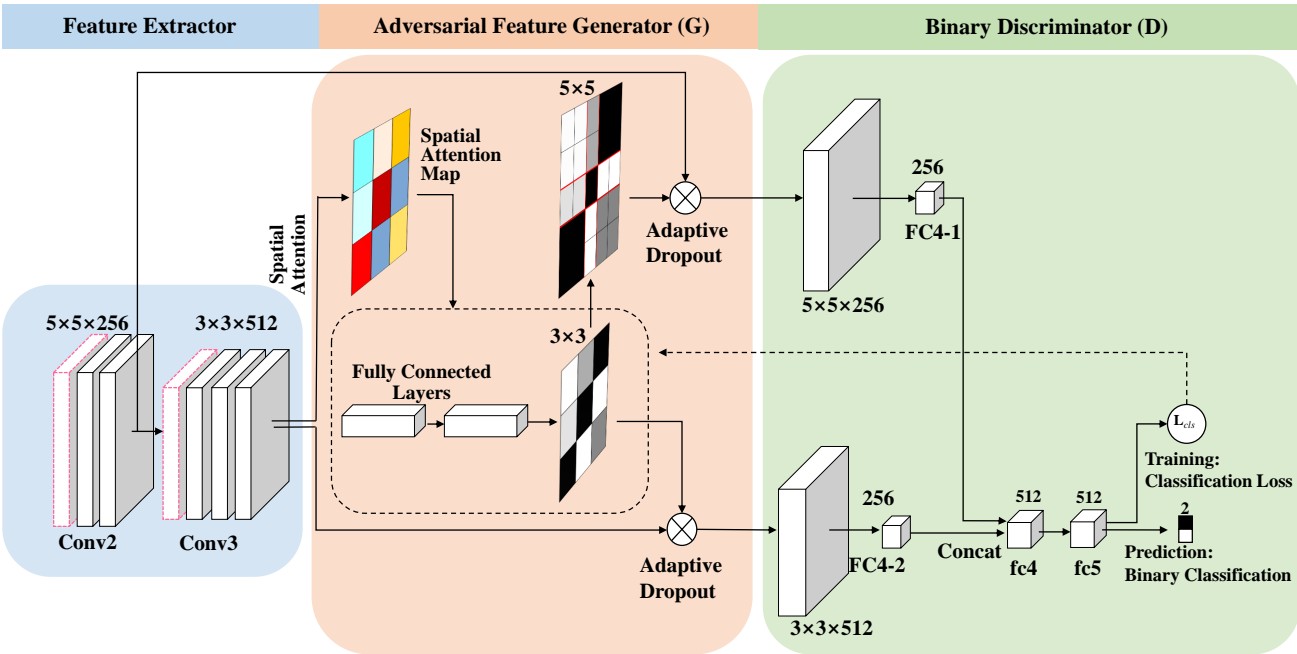

**Figure 2.** The architecture of AMGN-based object-tracking method.

### 3.2. Base Deep CNN and Tracking Network

Figure 3 shows the architecture of base deep CNN and tracking network. The first three convolution layers, Conv1-Conv3 from VGG-M, are used as the base deep CNN to extract the base deep features. The discriminant network takes over the fused features from Conv2 and Conv3 according to the form of fully connected layers and discriminates whether the feature belongs to the target. As we all know, lower-level CNN features have a higher spatial resolution to describe target locations, but show less semantic information, while higher-level CNN features are robust to target variations but with the absence of location information. For the purpose that the discriminator makes a better decision and the use of mask in AMGN makes up for much of the loss of features,we fuse Conv2 and Conv3 in the tracking network. In addition, in order to train the CBAM network's ability to recognize target robustness features independently, CBAM is placed after Conv3 for offline training and the parameters are retained.

### 3.3. Attention-Based Mask Generative Network

The attention mechanism can effectively focus eyes on areas of images that are discriminative to objects and backgrounds. Therefore, the human brain can devote more attention to these areas, obtain more details about the target, and suppress other useless information [52]. Attention weights acted on CNN feature maps also mark the most distinguishing feature that can assist the discriminator to make decisions, while the occluding of these features will always mislead the discriminator. However, combining the adversarial learning, the more these features are occluded, the more robust the discriminator can be.

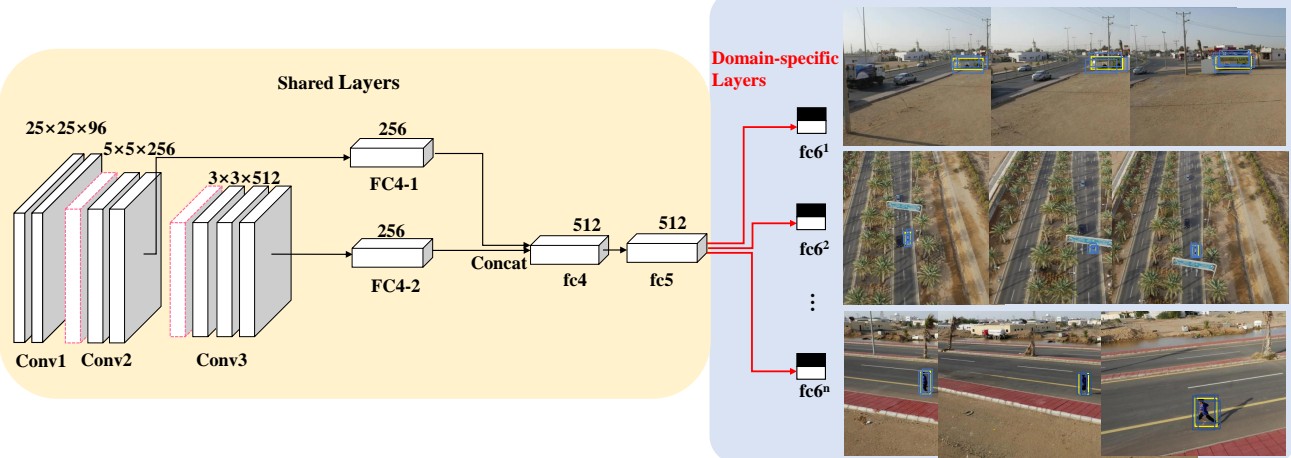

**Figure 3.** The architecture of base deep CNN and tracking network, consisting of shared layers and *n* branches of domain-specific layers.

In the proposed method, for all extracted positive sample feature maps, the result of the spatial attention module has the same size as $M^3$ of a single channel. $C^3$ is the output feature map of Conv3. Positions at which there is the maximum in the spatial attention weight matrices are set to zero, which compose the candidates' label of the mask. By selecting the template with the lowest classification score when it instructs $C^3$ to dropout features, it will be the final generation label *M*. In Figure 4, we give some examples of candidate labels generated based on spatial attention matrices.

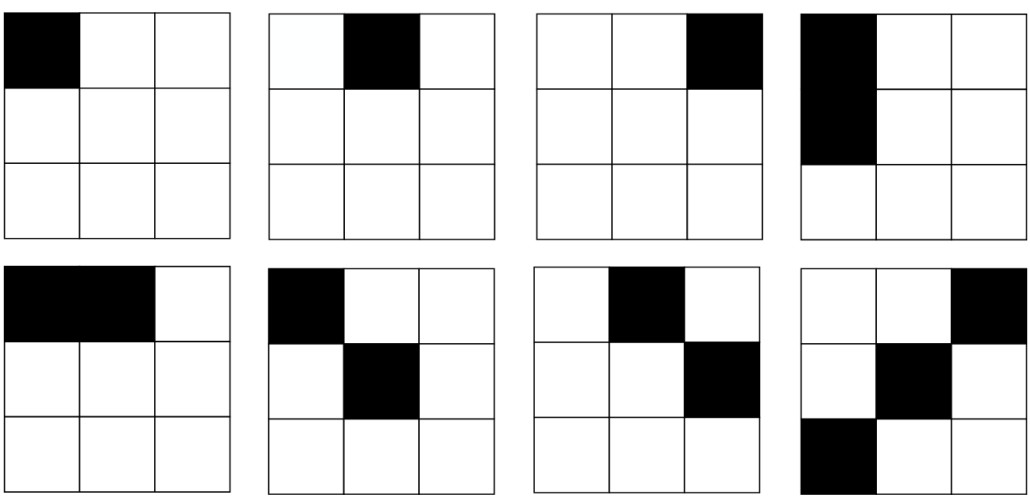

**Figure 4.** Examples of candidate labels generated based on spatial attention matrices.

Different input features will conduct different and continuous heatmaps under the guidance of the assigned label *M* with AMGN, which is composed of two fully connected layers. Here, mean squared error (MSE) is utilized to measure the difference between estimated generated masks and the assigned label. Given a feature map *C* with the size of $W \times H$, the MSE loss can be expressed as

$$L_{MSE} = \frac{1}{N} \sum_{j=1}^{N} \left( M_j - \hat{M}_j \right)^2,$$ (3)

where $\hat{M}$ and *M* denote the generated masks and assigned labels, respectively.

After the thresholding operation where we select the top $\frac{1}{3}$ pixels as 0 and others as 1, the generated masks become our attention-based masks. The dot product of the maskwith $C^3$ of multiple channels obtains the feature sample under occlusion, which is described by

$$C_M^3 = C^3 \cdot \hat{M}, \tag{4}$$

$$\hat{M} = G(C^3), \tag{5}$$

where $C_M^3$ is the output that the attention-based mask acts on $C^3$. $G$ represents the generation operation of the AMGN. $\hat{M}$ is the attention-based mask.

*3.4. Feature Fusion*

Since the size of $M$ is inconsistent with the size of $C^2$ in the feature graph after the second convolution, the mask needs to be processed. The weight values of rows and columns in $M$ are dot multiplied by the corresponding rows and columns in $C^2$ to obtain $C_M^2$, that is, $M^2$ with the same size of $C^2$ is the mask processed by $M$. Finally, the two-layer feature images processed by the fully connected layer are cascaded and sent to the final target classification branch. The values at $(r_A, c_A)$ in adjusted $H_A \times W_A$ mask $M^2$ are equal to the values at $(r_B, c_B)$ in base $H_B \times W_B$ mask $M$, where $\{(r_A, c_A) | \lfloor r_1 \rfloor + 1 \le r \le \lfloor r_2 \rfloor, \lfloor c_1 \rfloor + 1 \le c \le \lfloor c_2 \rfloor; r, s \in \mathbb{Z}\}$. The $r_1, r_2, c_1$ and $c_2$ are calculated by

$$r_1 = ((r_B - 1)H_A / H_B) + (1/2), \tag{6}$$

$$r_2 = (r_B H_A / H_B) + (1/2), \tag{7}$$

$$c_1 = ((c_B - 1)W_A / W_B) + (1/2), \tag{8}$$

$$c_2 = (c_B W_A / W_B) + (1/2). \tag{9}$$

Mask $M^2$ is directly obtained by the transformation of mask $M$. Finally, the object function of AMGN is defined as

$$\mathcal{L}_{AMGN} = \min_G \max_D \mathcal{L}_1 + \mathcal{L}_2 + \lambda \mathbb{E}_{(C^3, M) \sim P(C^3, M)} \left\| G(C^3) - M \right\|^2, \tag{10}$$

$$\mathcal{L}_1 = \mathbb{E}_{(C^3, C^2, M) \sim P(C^3, C^2, M)} [\log D(M \cdot C^3, f^{C^2})], \tag{11}$$

$$\mathcal{L}_2 = \mathbb{E}_{(C^3, C^2) \sim P(C^3, C^2)} [\log(1 - D(G(C^3) \cdot C^3, f^{C^2}))], \tag{12}$$

where $G$ represents the generative network, $D$ represents the discriminative network, and $M$ is the theoretically optimal mask matrix under the premise of a given feature map, which refers to the mask matrix that is most likely to make the $D$ error, while the goal of $G$ is to make it generate matrix $G(C^3)$, which is closest to the optimal matrix $M$, as the input is $C^3$. $f$ denotes the operation described in Equations (6)–(11) that adjust the size of the mask and perform it on the feature map $C^2$.

In the process of online training, $G$ is fixed at first. $\max_D \mathbb{E}_{(C^3, C^2, M) \sim P(C^3, C^2, M)} [\log D(M \cdot C^3, f^{C^2})] + \mathbb{E}_{(C^3, C^2) \sim P(C^3, C^2)} [\log(1 - D(G(C^3) \cdot C^3), f^{C^2})]$ has the requirements that increase $D(M \cdot C^3, f^{C^2})$ and decrease $1 - D(G(C^3) \cdot C^3, f^{C^2})$ at the same time, which means the demand toward $D$ to distinguish the difference between $M \cdot C^3$ and $G(C^3) \cdot C^3$. Then, $D$ is fixed and $G$ is optimized.

To achieve $\min_G \mathbb{E}_{C^3 \sim P(C^3)} [\log(1 - D(G(C^3) \cdot C^3))] + \lambda \mathbb{E}_{(C^3, M) \sim P(C^3, M)} \left\| G(C^3) - M \right\|^2$, $D(G(C^3) \cdot C^3)$ should be increased, while $\mathbb{E}_{(C^3, M) \sim P(C^3, M)} \left\| G(C^3) - M \right\|^2$ should be decreased, and also $G$ should be trained to make $G(C^3)$ close to the theoretical optimal mask matrix $M$. In this process, generative network $G$ and discriminant network $D$ play games with each other and evolve alternately to form a generative adversarial network structure. It should be noted that the optimal mask matrix M is the most error-prone matrix of discriminant network $D$, and discriminant network $D$ treats $M \cdot C^3$ as a positive sample.

*3.5. Tracking Process*

The tracking process of our proposed AMGN tracker includes three parts: model initialization, online detection, and online model update.

(1)   Model Initialization: The base CNN is initialized by VGG-M [53] trained in the classification task from ImageNet. The parameters in Conv1-Conv3 of the base CNN are fixed and the others are initiated according to offline pre-training by multidomain learning, which is fine-tuned online.

(2)   Online Detection: Generated multiple candidate boxes on the first frame of the tracking sequence or previous frame and its predicted target position are sampled by base CNN in each and fed into the tracking network to obtain probability scores.

(3)   Online Model Update: According to the target position given in the first frame and the predicted target position in other frames, we generate multiple candidate boxes around them and assign two-category labels divided by intersection-over-union (IoU) scores. The labeled samples are used to jointly train AMGN (as the generator $G$ of GAN) and tracker (as the discriminator $D$ of GAN) to complete the adversarial processes. AMGN produces the attention-based mask firstly as the $C^3$ input, the mask adjustment process then meets the size of the mask to $C^2$ and obtains the fused occluded feature maps. With the label unchanged, $D$ is studied through supervised learning. After training, $D$ will suffice for identifying the target features occluded. In return, $D$ guides $G$ to generate more difficult masks for $D$.

## 4. Experiments

To evaluate the performance of the proposed AMGN-based tracker, we perform extensive experiments in terms of accuracy and robustness.

At present, numerous VOT benchmark datasets have been introduced to provide fair and standardized evaluations of object-tracking algorithms. For UAV-specific tracking, commonly used datasets include UAV123 [54], DTB70 [55], UAVDT [56], UAV20L [54], VisDrone2018 [57], etc. For generic tracking, OTB2015 [58], VOT2016 [59], GOT10k [60], FaceTracking [61], etc., have been widely used for evaluation. In order to fully verify the performance of the proposed method in UAV-specific tracking, we select three datasets, including UAV123, DTB70 and UAVDT, for experimental verification. In addition, considering the application potential of our method in generic object tracking, we also conduct comparative experiments on VOT2016.

*4.1. Implementation*

In this work, the first three convolution layers are from VGG-M [53] trained on ImageNet and always fixed in the process of the online tracking process. After Conv3, a CBAM that is also trained offline is used to obtain spatial attention maps. Spatial attention maps with the same size with $C^3$ ($3 \times 3$) are adjusted to the binary as candidate masks. We train $D$ first by applying candidate masks independently to each fused feature and choosing the one with the lowest classification score. Here, we alert $3 \times 3$ masks to $5 \times 5$ and meet the size of $C^2$. The fused feature is from $C^2$ and $C^3$. Then, the trained $D$ guides $G$ to generate masks like the label but composed of numbers between 0 and 1. During the adversarial learning process, the SGD solver is iteratively applied to $G$ and $D$. The learning rate is set to $10^{-3}$ and $10^{-4}$, respectively. We update both networks every 10 frames using 10 iterations. The whole experiment is performed on a PC with an i7-8700 CPU and NVIDIA GeForce GTX 1660 Ti GPU.

*4.2. Evaluation on UAV123*

UAV123 [54] contains 123 UAV videos with 12 challenging attributes, including illumination variation, scale variation, full occlusion, partial occlusion, camera motion, etc. In this paper, we follow with interest the overcoming of tracking drift under full occlusion and partial occlusion. We use the one-pass evaluation (OPE) metrics to measure the tracking performance. The precision plot computes the percentages of frames whose estimated

locations lie in a given threshold distance to ground truth centers. The typical threshold distance is 20 pixels. The success plot is set to measure the overlap score (OS) between the ground truth and the bounding box resulting from the tracker. Afterward, a frame whose OS is larger than a certain threshold is termed a successful frame, and the success rates under different thresholds constitute a success plot. The general threshold is set to 0.5.

We compare an AMGN-based tracker with 21 state-of-the-art trackers composed by CF-based and CNN-based trackers, including ECO [62], ECO-HC [62], SiamRPN [11], C-COT [63], STRCF [64], DeepSTRCF [64], TADT [65], SRDCF [66], SRDCFdecon [67], SAMF [68], Staple [69], Staple_CA [70], KCF [71], KCC [72], DSST [73], UDT [74], UDT-plus [74], BACF [75], CSRDCF [76], MEEM [77], and MOSSE [5]. We evaluate all the trackers on 123 video sequences through OPE with distance precision and overlap success metrics.

Figure 5 shows the success and precision plots for the top 10 of the 21 comparison trackers. The values listed in the legends are the AUCs of the success rates and the 20-pixel distance precision scores, respectively. It is evident that our AMGN-based tracker performs well compared to other state-of-the-art trackers, with a leading precision score of 0.779.

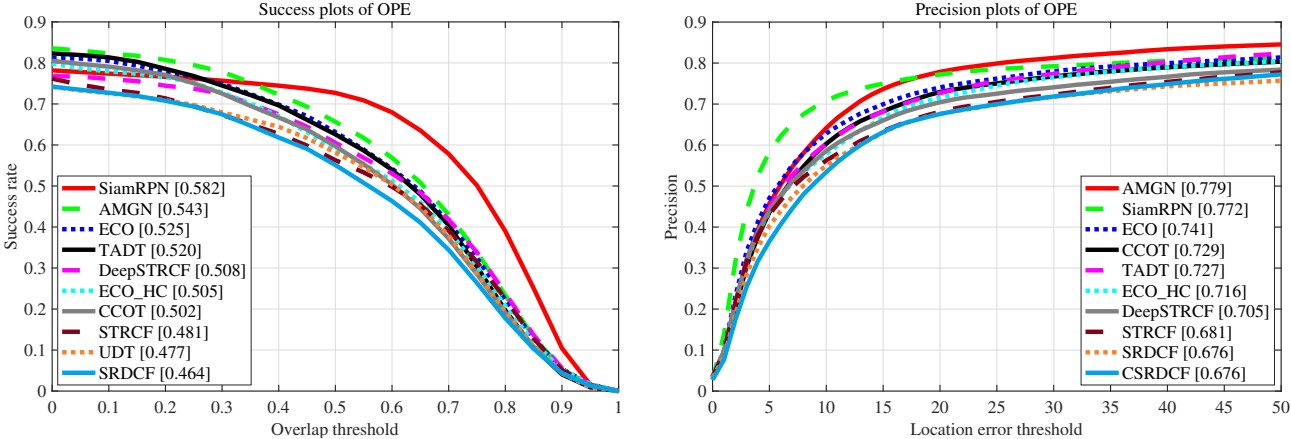

**Figure 5.** Success and precision plots on the UAV123 dataset using one-pass evaluation.

In Figure 6, we further show the success and precision plots under the two attributes: partial occlusion and full occlusion. The results show that the AMGN-based tracker achieves the best performance in handling occlusion challenges compared to the comparison trackers. The AUCs of the success plots under the two attributes lead the runner-up by 1.4% and 2.3%, respectively, and the precision scores lead the runner-up by 6.9% and 8.7%, respectively.

### 4.3. Evaluation on DTB70

We also evaluate 70 sequences from the DTB70 dataset [55] with 12 attributes, including scale variation, occlusion, deformation, fast camera motion, similar objects around, etc. Based on the same metrics as UAV123, we compare our proposed algorithm with 24 other state-of-the-art trackers: ECO [62], ECO-HC [62], C-COT [63], BACF [75], CoKCF [78], STRCF [64], DeepSTRCF [64], TADT [63], SRDCF [66], SRDCFdecon [67], SAMF [68], SAMF_CA [70], Staple [69], Staple_CA[70], KCF [71], KCC [72], MOSSE [5], MCCT [79], MCCT_H [79], MCPF [80], DSST [73], fDSST [81], UDT [74], and IBCCF [82].

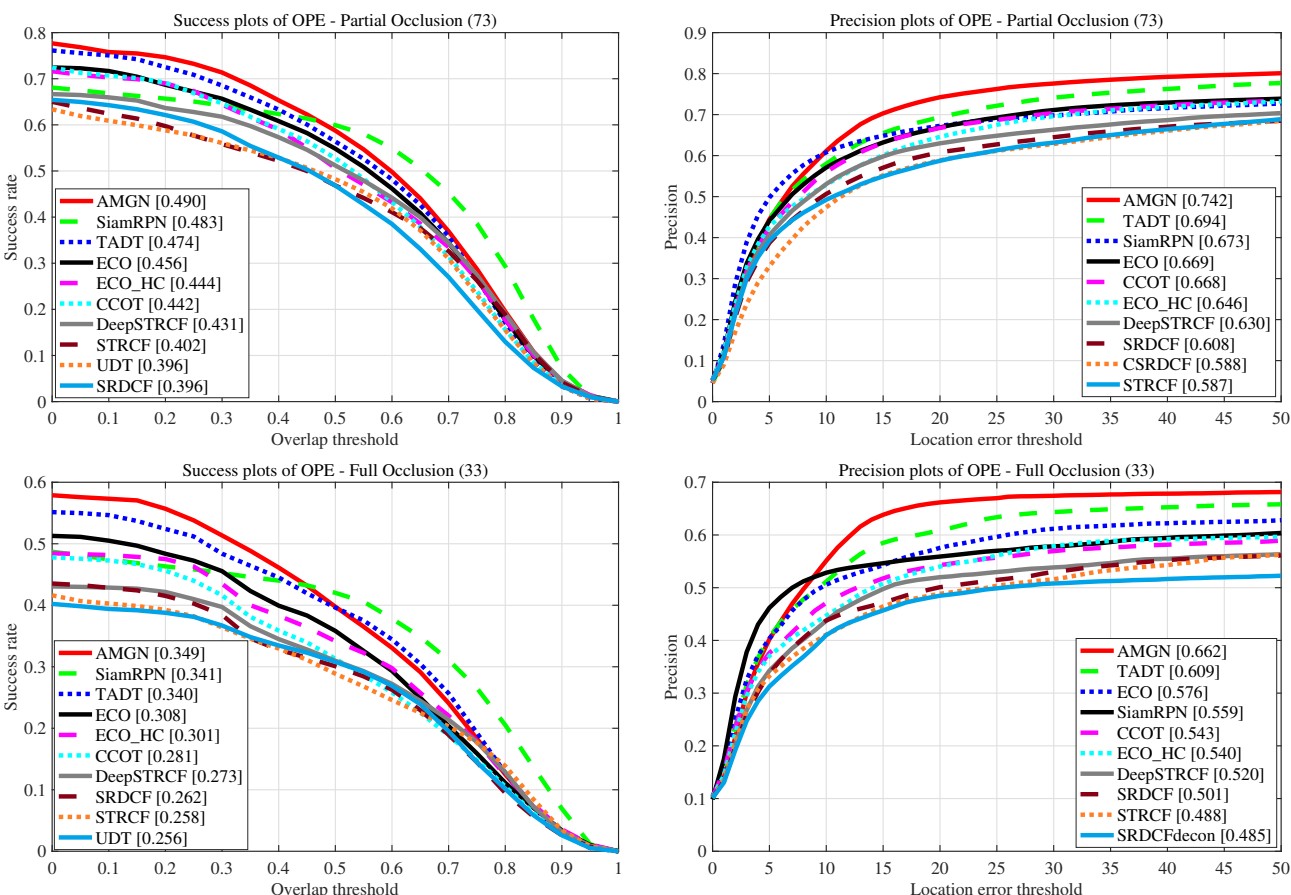

**Figure 6.** Success and precision plots on the UAV123 dataset using one-pass evaluation over tracking challenge occlusion.

Figure 7 shows the success and precision plots for the top 10 of the 21 comparison trackers. Similarly, the values listed in the legends are the AUCs of the success rates and the 20-pixel distance precision scores, respectively. Our AMGN-based tracker achieves the best AUC of 0.539 and the best precision score of 0.788 for overall videos.

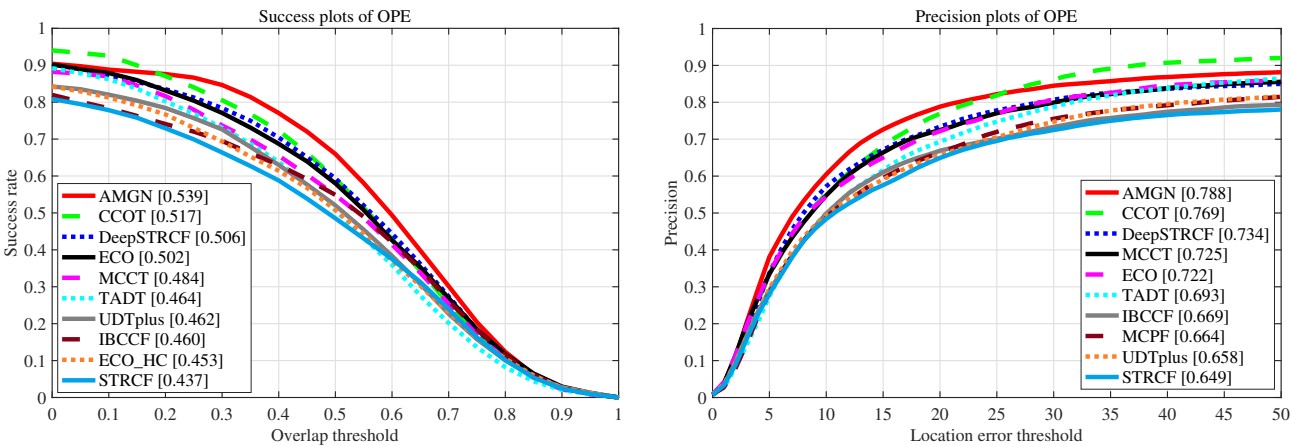

**Figure 7.** Success and precision plots on the DTB70 dataset using one-pass evaluation.

In Figure 8, we present the success and precision plots under the attribute, deformation. The results show that the AMGN-based tracker also achieves the best performance in handling deformation challenges. The AUC of the success plot leads the runner-up by 2.3%, and the precision score leads the runner-up by 9.1% .

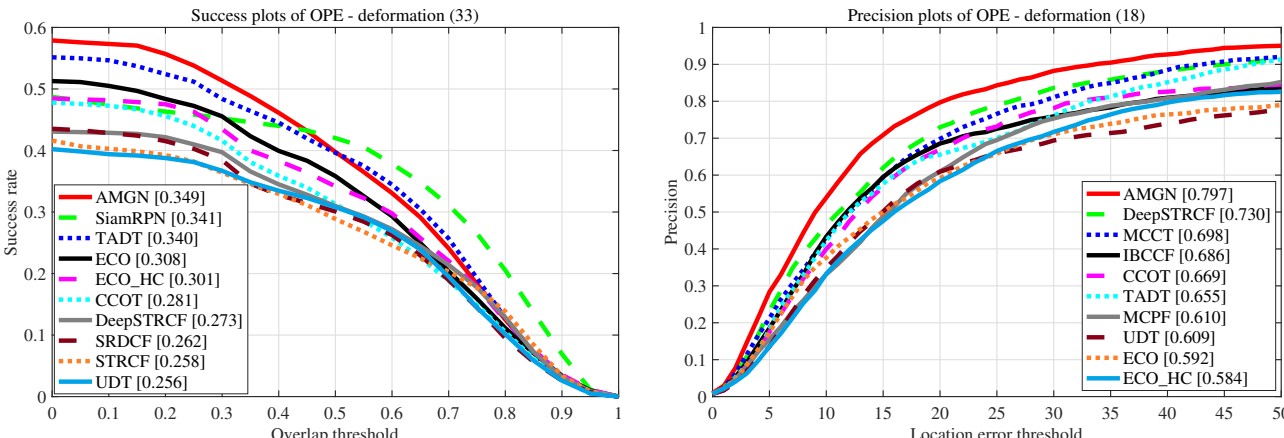

**Figure 8.** Success and precision plots on the DTB70 dataset using one-pass evaluation over tracking challenge Deformation.

### 4.4. Evaluation on UAVDT

For a more comprehensive evaluation, the proposed AMGN-based tracker is additionally compared with another 13 state-of-the-art trackers on UAVDT benchmark, including SiamFC [83], ECO [62], MDNet [34], CREST [84], C-COT [63], Staple_CA [70], SRDCFdecon [67], KCF [71], CFNet [85], MCPF [80], SRDCF [66], UDT [74], and SINT [22]. In the single object tracking task, UAVDT is composed of 50 aerial tracking videos with 8 attributes, including background clutter (BC), camera rotation (CR), object rotation (OR), small object (SO), illumination variation (IV), object blur (OB), scale variation (SV) and large occlusion (LO). Among these attributes, large occlusion is what we are concerned with. Similar to UVA123 and DTB70, we also conduct the evaluation through OPE with overlap success metrics and distance precision.

The success and precision plots for the top 10 of the 13 comparison trackers on the UAVDT dataset are presented in Figure 9. The proposed tracker exhibits the leading AUC of the success plot of 0.528 and the leading precision score of 0.771 for overall sequences.

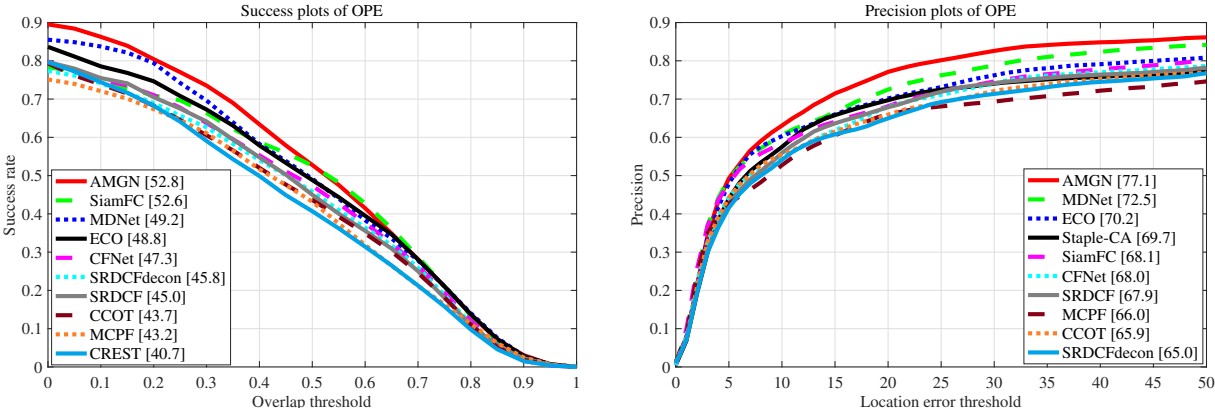

**Figure 9.** Success and precision plots on the UAVDT dataset using one-pass evaluation.

Meanwhile, Figure 10 presents the success and precision plots under the attribute, large occlusion (LO). The results show that our proposed AMGN-based tracker outperforms the second-best tracker 4% and 6.4% in AUC regarding the success plot and precision score, respectively.

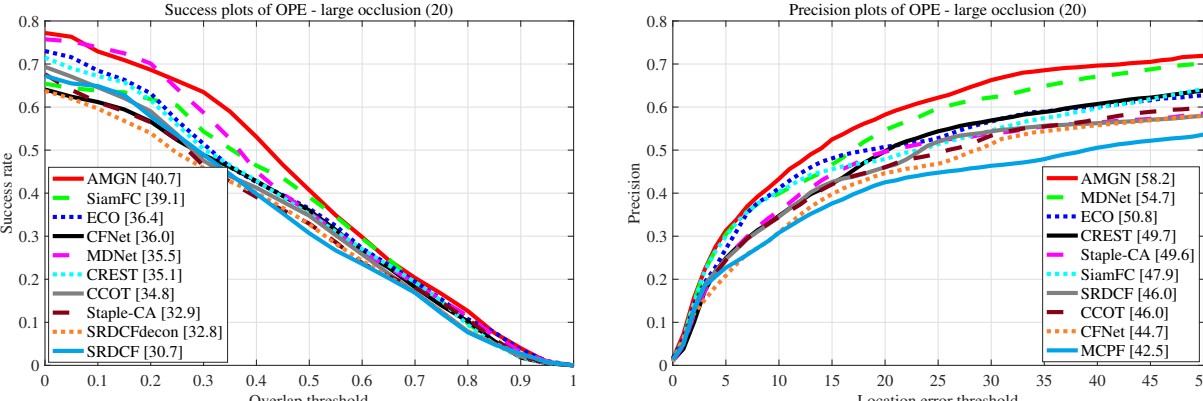

**Figure 10.** Success and precision plots on the UAVDT dataset using one-pass evaluation over tracking challenge large occlusion.

*4.5. Evaluation on VOT2016*

We also conduct a supervised evaluation on 60 sequences from the VOT2016 dataset [59]. Based on the VOT challenge protocol, trackers are re-initialized once a tracking drift is detected [86]. We use three metrics to evaluate the performance: expected average overlap (EAO) on short-term sequences, average overlap during the periods of successful tracking (accuracy), and the average number of failures during tracking (robustness). In Table 1, we compare our proposed algorithm with six other state-of-the-art trackers: ECO [62], CCOT [63], VITAL [10], MDNet [34], CREST [84], and Staple [69]. Our AMGN-based tracker achieves the best accuracy and robustness and the second EAO, which is sufficient to show its availability.

**Table 1.** Quantitative comparison results on the VOT2016 dataset. Values in red and green indicate the best and the second-best performance, respectively.

|  | ECO | CCOT | VITAL | MDNet | CREST | Staple | AMGN |
|---|---|---|---|---|---|---|---|
| EAO ↑ | 0.374 | 0.331 | 0.323 | 0.257 | 0.283 | 0.295 | 0.340 |
| Accuracy ↑ | 0.54 | 0.52 | 0.55 | 0.57 | 0.51 | 0.54 | 0.576 |
| Robustness ↓ | 0.72 | 0.85 | 0.98 | 1.20 | 1.08 | 0.35 | 0.191 |

*4.6. Ablation Studies*

4.6.1. Effectiveness of Attention Module and Adversarial Learning

In the AMGN-based tracker, we align the attention mechanism to the diversified hard positive samples as an AMGN module and train the classifier to overcome tracking drift adversarially. To validate the effectiveness of our AMGN module, we make experiments on the baseline tracker, baseline tracker with CBAM after the Conv3, and our proposed tracker. In addition, we add CBAM after $C^3$ to prove that an attention module in the high-level convolution layer can effectively avoid tracking drift, for it can benefit valid features and restrain the others. On the other hand, when we reversely drop out the target features of the focused parts of attention in our proposed method, the network concentrates attention on the whole target with more robustness characteristics, causing further improvement of the tracking effect. Figure 11 shows the ablation studies results on the DTB70 dataset. We observe that joint CBAM and adversarial learning produces significant improvements overall in both occlusion-tagged and deformation-tagged sequences. For example, compared with the baseline, the proposed AMGN-based tracker improves the precision scores by 18.4%, 11.4%, and 19.8% overall, in occlusion-tagged and deformation-tagged sequences, respectively.

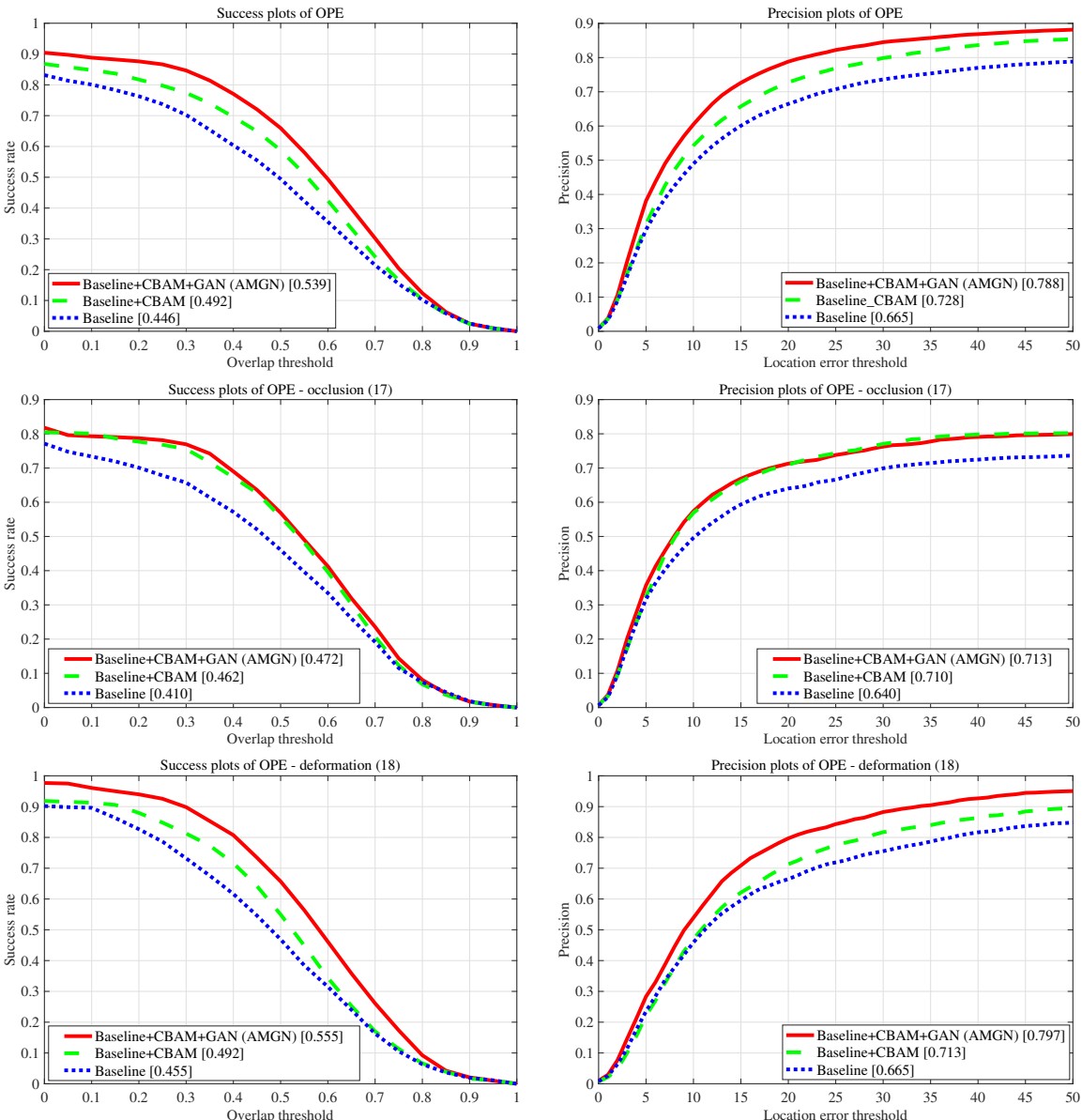

**Figure 11.** Ablation studies results on attention module and adversarial learning on the DTB70 dataset.

### 4.6.2. Effectiveness of Feature Fusion

To verify the effectiveness of our designed feature fusion method, we conduct further ablation studies on it. Since the original intention of the feature fusion method is to compensate for the excessive dropout of features that may be caused by the masks, in this experiment, we use the final formed AMGN-based tracker as the baseline and compare it with the configuration after only removing the feature fusion module. The results are shown in Figure 12. It can be seen that after removing the feature fusion method, the overall performance of the tracker and in both cases of occlusion and deformation has a significant decline, indicating the effectiveness of the feature fusion method. Specifically, after introducing the feature fusion method, the precision scores in the overall, occlusion and deformation sequences are improved by 10.8%, 5.6%, and 7.4%, respectively.

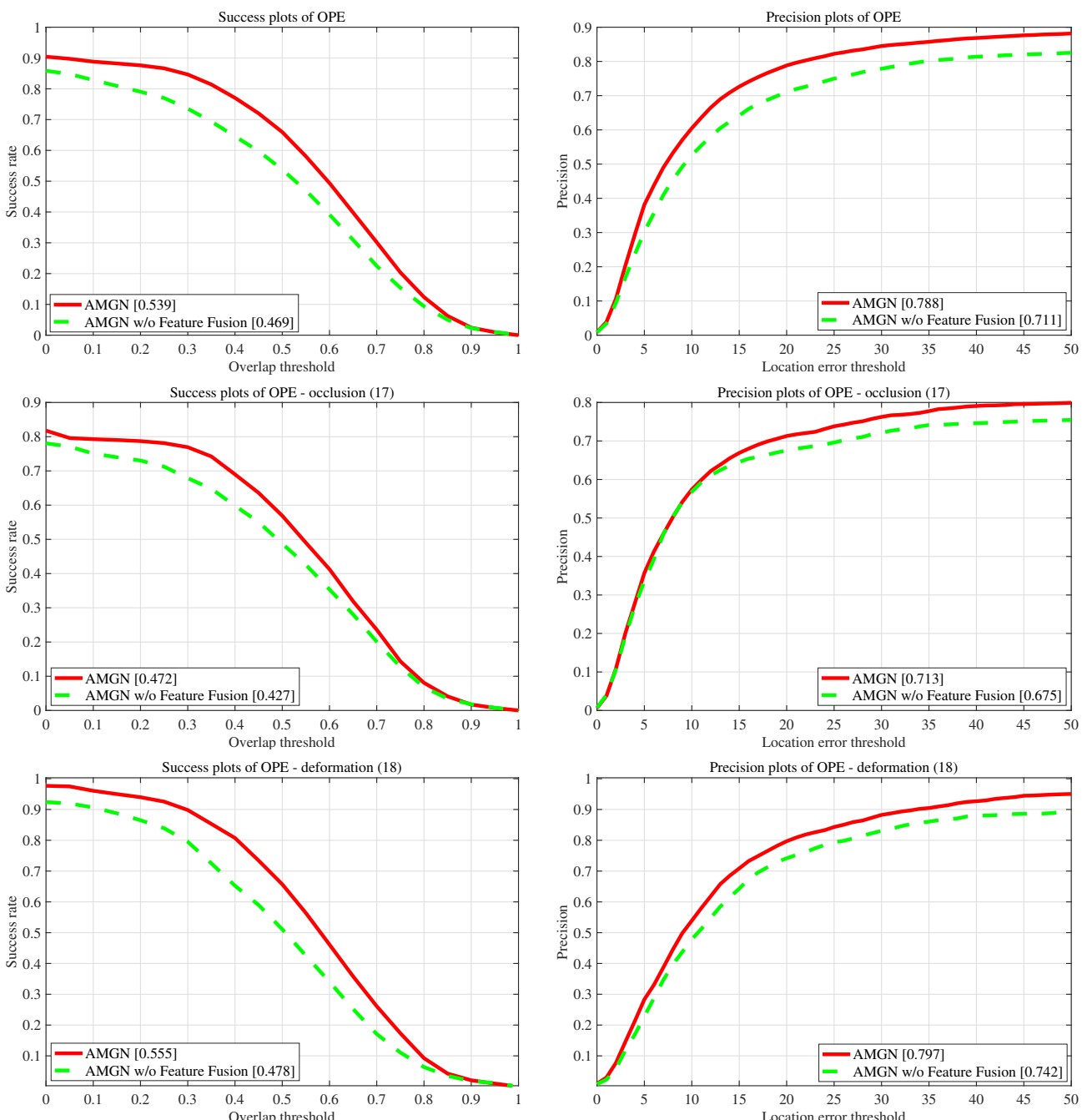

**Figure 12.** Ablation studies results on feature fusion on the DTB70 dataset.

### 4.7. Qualitative Evaluation

Figure 13 qualitatively compares the results of the top-performing trackers: ECO, CCOT, SiamRPN, SRDCFdecon, STRCF, BACF, and the proposed AMGN-based tracker on 13 challenging sequences. We choose six sequences from UAV123 with the attributes of full occlusion and partial occlusion, and another sequence occurs, deformation. In most sequences, SiamRPN, STRCF, and SRDCFdecon fail to locate the target with weak performance once there is occlusion. ECO and CCOT, despite unit CNN with correlation filtering and receiving richer feature representation, also lead to track failure when the target is fully blocked or there is interference of similar objects, as they do not take full advantage of the end-to-end deep architecture. The AMGN-based tracker keeps the best success rate under extreme conditions, especially in almost complete occlusion, and occlusion reappears because of our attention mechanism. It is noticed that the SimaRPN- and AMGN-based

trackers show higher tracking precision under deformation. To verify the anti-deformation ability of our proposed tracker, we choose seven sequences from DTB70 that have bigger objects and obvious deformation from the qualitative results, even under the influence of strong deformation, clutter background, etc.

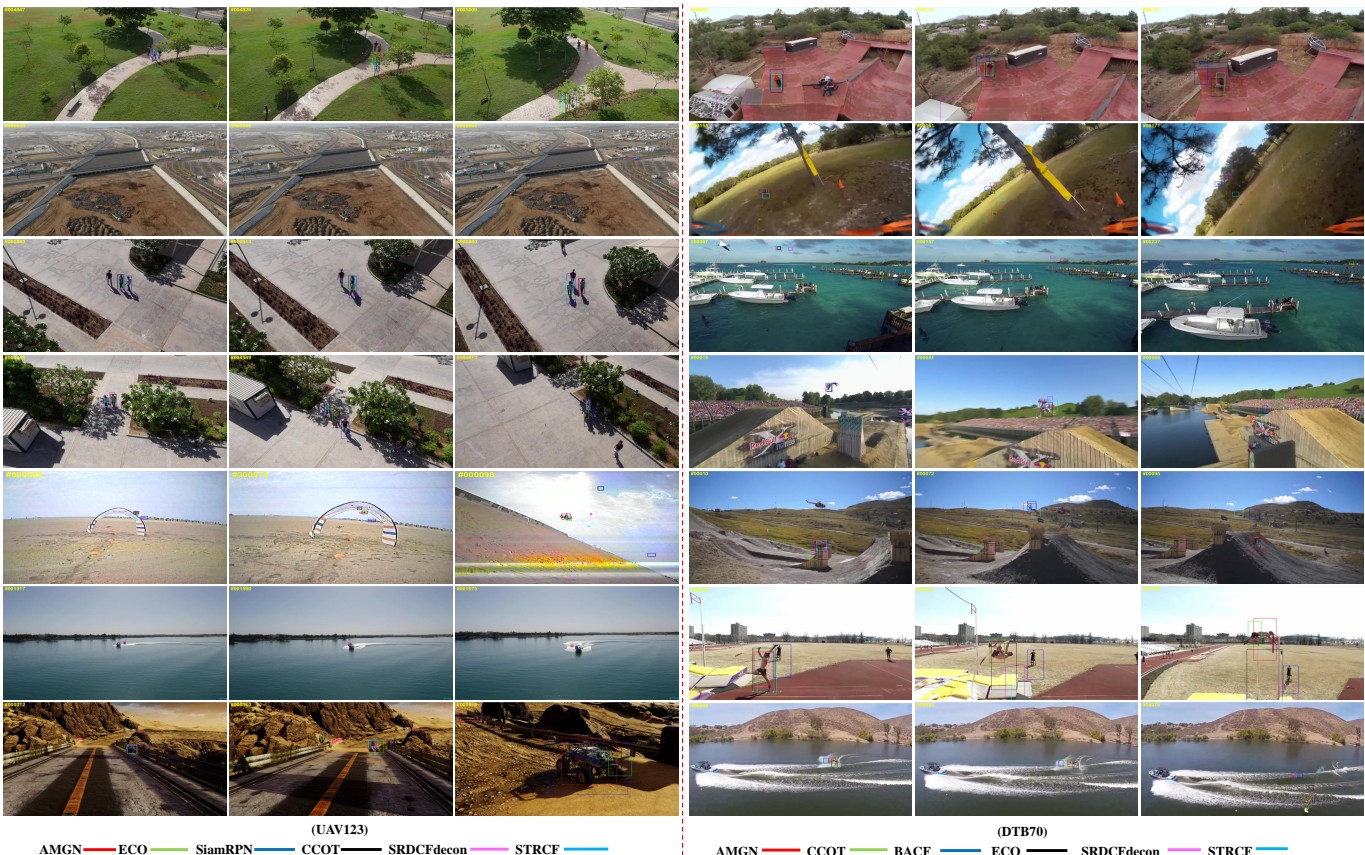

**Figure 13.** Qualitative results of the comparison tracker on challenging sequences from UAV123 and DTB70.

## 5. Conclusions

In this paper, we propose an AMGN-based tracker that leverages adversarial learning to enhance the tracker's resilience to occlusion and deformation. After extracting the deep features of the candidate target region through a base CNN, we first develop an attention-based mask generative network (AMGN), which adopts the attention mechanism to calculate the attention map of the deep features of the candidate target regions, and generates a series of masks according to the attention map. These masks are multiplied with deep features to simulate target occlusion and deformation in feature space. Then, to avoid over-dropping of target features by masks, we design a feature fusion method that incorporates shallower-layer features into deeper-layer features processed by masks, thus avoiding extreme cases of tracking drift due to excessive feature loss. After the above processing, the hard positive samples focusing on target occlusion and deformation are supplemented. Finally, we treat the tracker as the discriminator of GAN and use these hard positive samples for adversarial learning, thereby improving the tracker's ability to deal with occlusion and deformation. Comparative experiments show that our AMGN-based tracker achieves the highest AUC of 0.490 and 0.349, and the highest accuracy scores of 0.742 and 0.662 on the UAV123 benchmark with partial and full occlusion attributes, respectively. On the DTB70 benchmark with the deformation attribute, it achieves the highest AUC of 0.555 and the highest precision score of 0.797. On the UAVDT benchmark with the large

occlusion attribute, it achieves the highest AUC of 0.407 and the highest precision score of 0.582.

Although the effectiveness of the proposed method is validated on several datasets, there are still some limitations. On the one hand, in order to improve the efficiency of the mask, we use the attention mechanism to generate the masks instead of doing so randomly, which will increase the computational complexity of the tracker; on the other hand, the method proposed in this paper focuses on considering the accurate target tracking under occlusion and deformation, so the overall performance improvement is not obvious. Further research will mainly focus on reducing the high computational complexity in terms of time and space, improving the flexibility of applying various basic CNNs, and considering more different challenging cases, thereby further improving the overall performance of the tracker.

**Author Contributions:** Conceptualization, Y.B.; methodology, Y.B. and Y.Z. (Yufei Zhao); software, Y.B., X.W., Z.Z. and Y.H.; validation, Z.Z. and Y.H.; formal analysis, Y.B.; investigation, Y.Z. (Ya Zhou) and X.Y.; resources, Y.S.; data curation, Y.B.; writing—original draft preparation, Y.B.; writing—review and editing, X.W., Y.H. and Y.B.; visualization, Y.H.; supervision, Y.S., Y.Z. (Yufei Zhao) and Q.H.; project administration, Y.S.; funding acquisition, Y.S. All authors have read and agreed to the published version of the manuscript.

**Funding:** This research was funded by the National Natural Science Foundation of China under Grant number 81671787.

**Data Availability Statement:** The current research was conducted on publicly available datasets.

**Acknowledgments:** The authors would like to thank all the colleagues who generously provided their image dataset with the ground truth. The authors would also like to thank the anonymous reviewers for their very competent comments and helpful suggestions.

**Conflicts of Interest:** The author declares no conflict of interest.

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
