# Peer review of "Occlusion and Deformation Handling Visual Tracking for UAV via Attention-Based Mask Generative Network"

_remotesensing, doi:10.3390/rs14194756_

Round 1

Reviewer 2 Report

Comments on the paper “

Occlusion and Deformation Handling Visual Tracking for UAV via Attention-based Mask Generative Network

Context comments

We authors state that “We proposed an attention-based mask generative method, which combines the mask generation network with the attention mechanism to adaptively form a simulation method for target occlusion and deformation in the feature space“. This novelty aspect must be explicitly discussed in the paper. More details should be furnished.

Since the proposed approach is oriented to improve anti-occlusion and anti-deformation abilities, the results of the numerical experiments must be presented considering these aspects. More details should be furnished.

What influence provide the proposed feature fuse technique?  What are the advantages and shortcomings of this proposal in real applications? More details should be furnished.

 The results of the application of the proposed feature fuse must also be presented and discussed. More details should be furnished.

The conclusions are quite trivial and must incorporate the obtained results due to the novel proposals presented in the manuscript.

Format comments

The same legend must be used in all figures.

Reviewer 3 Report

This paper presents an attention-based mask generative model to address two important issues, occlusion and deformation, in UAV tracking. The idea is interesting to align attention and adversarial learning and experiments show its effectiveness on two datasets. Overall, this is a good paper and the following concerns should be addressed.

1. The difference between this work and the adversarial learning tracking [a], attribute-based tracking [b] and state-based tracking [c] should be discussed in the Introduction or Related work. [b] and [c] are also designed to handle occlusion and deformation.

[a] VITAL: VIsual Tracking via Adversarial Learning

[b] Learning attribute-specific representations for visual tracking

[c] Robust visual tracking via scale-and-state-awareness

2. The proposed method seems to be able to apply to general object tracking. It would be better to evaluate on other tracking datasets which also challenged by occlusion and deformation, such as UAVDT[a] and FaceTracking[b]

[a]UAVDT: The unmanned aerial vehicle benchmark: Object detection and tracking

You can compare with public results in: Visdrone-sot2018: The vision meets drone single-object tracking challenge results

[b]FaceTracking: Siamese local and global networks for robust face tracking

3. Several important general visual tracking strategies are missing in related work, such as

[a] ATOM: Accurate Tracking by Overlap Maximization

[b] Release the power of online-training for robust visual tracking

[c] Probabilistic Regression for Visual Tracking

[d] Structure-aware multi-object discovery for weakly supervised tracking

Reviewer 4 Report

Dear Authors,

   Thanks a lot for your manuscript submission to MDPI Journal of Remote Sensing. This paper proposed an attention-based mask generative network (AMGN) on handling visual tracking related topics (occlusion handling, deformative issues, precision and success plots, etc), the approach is joint model on attention mechanism and adversarial learning, the architecture is base deep learning and their variations on visual tracking. This paper has mild contributions and the overall quality is acceptable. Hence, I recommend it as acceptance with minor revision. There are some problems needs to be fixed, the suggested edits are listed below.

   Major problematic issues are specified as follows:

   a) Abstract: It is quite short in length and acceptable in organization. Wile some redundant statements, i.e., several sentences in Lines 1-6 need to be reshaped. The last two sentences on experimental results need to be more clear and specific, adding crucial quantitative conclusion is preferred. About 150-200 words are expected in the revised version. Thanks a lot!

   b) Introduction: I think the first and second paragraphs may need a major re-write. Citing more than 10 articles (i.e., [3-14] in Line 29 and [15-27] in Line 31) all at once makes too little sense. I suggest you expanding some of the narrations and replace some gaps with your specific understandings on the weakness of prior and current research study. Besides, any study within the latest two years 2020-2022, should be clearly addressed. Besides, while this section presented a short summary on major contributions on this set of research work, specific claims should be updated in Lines 75-85 (with respect to each manifold). In Lines 92-94, the last part specifies the remainder of this paper on the related sections, looks a bit too generic. Please update. Thanks!

   c) Section 2: Related work. The current section basically looks fine, it is a short summary of several aspects on learning. I think the authors may consider adding some of the transition between subsections be cohesively connected. Avoid using "hard brake" to end up a paragraph. Thanks a lot! 

   d) Figures and Tables: The positioning of Fig. 9 and Fig. 10 need to be exchanged. I suggest the authors reshaping the Pages 12-13. The legends and annotations should be uniform and clearly visible. Please update these figures. Besides, this paper lacks to present any table, which might be a main shortcoming by specified by other reviewers. The title of each figure need to be applied with middle alignment. Also, be sure to avoid crossing over two adjacent pages (as the current section did well for all these figures). 

   e) Approach: The performance evaluation suggested that AMGN are superior to most of the other state-of-the art schemes, but to be frank, the improvements are not very explicit. I think the authors need to explain this issue and check whether your approach is better than the best visual trackers in Years 2020-2022, or at least fill in the potential holes in the conclusion section. Otherwise, Please consider improving the idea, or at least conduct some variation on this model. Think about this and we appreciate that.

   f) Discussion: I think this paper missed a paragraph on discussing the limitations of your study, the pros and cons of your proposed approach, and parallel comparsion to any latest publications (if reporting even better outcome other than your approach). Please consider filling this role if applicable. You may shift the rest results in Subsections 4.4 ~ 4.5 to the discussion section, and apply it with decent statements..

   g) Conclusion: Current section has main summary while it is absent from keynote quantitative results; also, part of your prospective study is missing. I suggest the authors adding a second paragraph (on future work) with a summary of research challenges and the orientations of your subsequent investigations, etc. Please just consider improving this crucial section. Thanks very much!

   h) References: a few obvious problems must be fixed. (i) Use abbreviated citation style. Please apply the required changes after the title of cited journals (MDPI does not require citing the issue number); (ii) Some latest publications in Years 2020-2022 parallel to your study area, are expected to be added, including the current state-of-the arts on UAV-based visual tracking and occlusion handling. Reference to wide-area aerial datasets are expected, both hybrid models and recent deep learning based approaches are supposed for citing. (iii) When citing conference proceedings, do please supplement the missed period, location and other information, i.e., Refs. [6], [10-15], [17], [20], [24-27], [30], [32-35], [50], [57], [58], [60], [61] and [65].

   Minor issues recommended for updates in your revised version: 

   a) This paper contains a few typos and capital letter issues (on the title of subsection, and title on some of the cited journals), you may need to carefully proofread the current version and fix any possible problems. 

   b) Literal quality of English can be further improved. I may suggest the co-authors inviting a native English speaker to polish this research article, which may include proofreading and grammatical checks in your updates. 

   c) If you are not using Latex, I suggest to avoid hyphenating a word which may cross over two adjacent lines. There are minor problens related to this in this content. MS word file of MDPI online template has the options to adjust that. Thanks a lot!

   d) The legends at left part of Figs. 8-9 should be adjusted a little bit to avoid overlapping the subfigures.  

   e) Meanwhile, Fix the related linespacing, font-type and size issues. Be consistently uniform with the required styles as specified by MDPI. 

   Once again, thank you, best of luck for your paper edits. Stay well and take care!

With warm regards,

Yours faithfully,

Round 2

Reviewer 2 Report

The manuscript can be published.

Author Response

Thank you for your affirmation. We appreciate your pertinent comments and valuable suggestions, which have helped us improve the manuscript.